# Recent Advances in Peptidoglycan Synthesis and Regulation in Bacteria

**DOI:** 10.3390/biom13050720

**Published:** 2023-04-22

**Authors:** Anne Galinier, Clémentine Delan-Forino, Elodie Foulquier, Hakima Lakhal, Frédérique Pompeo

**Affiliations:** Laboratoire de Chimie Bactérienne, UMR 7283, Institut de Microbiologie de la Méditerranée, CNRS/Aix-Marseille Univ, 31 Chemin Joseph Aiguier, 13009 Marseille, France

**Keywords:** peptidoglycan, PBP, SEDS, hydrolases

## Abstract

Bacteria must synthesize their cell wall and membrane during their cell cycle, with peptidoglycan being the primary component of the cell wall in most bacteria. Peptidoglycan is a three-dimensional polymer that enables bacteria to resist cytoplasmic osmotic pressure, maintain their cell shape and protect themselves from environmental threats. Numerous antibiotics that are currently used target enzymes involved in the synthesis of the cell wall, particularly peptidoglycan synthases. In this review, we highlight recent progress in our understanding of peptidoglycan synthesis, remodeling, repair, and regulation in two model bacteria: the Gram-negative *Escherichia coli* and the Gram-positive *Bacillus subtilis*. By summarizing the latest findings in this field, we hope to provide a comprehensive overview of peptidoglycan biology, which is critical for our understanding of bacterial adaptation and antibiotic resistance.

## 1. Introduction

Louis Pasteur is one of the most famous scientists whose scientific rivalry with Robert Koch paved the way for the study of microorganisms and the development of strategies to combat pathogenic bacteria. Since then, many advancements have been made in microbiology, including the accidental discovery of the antibacterial effects of penicillin by Alexander Fleming in 1928. Fourteen years later, Ernst Boris Chain and Howard Walter Florey used their findings to develop the first antibiotic, penicillin. We now know that it targets penicillin-binding proteins (PBPs) involved in the synthesis of peptidoglycan (PG), a crucial component of the bacterial cell wall.

PG plays a vital role in protecting bacteria from external stress and cytoplasmic pressure while maintaining its morphology [1,2]. This polymer comprises glycan chains crosslinked by short peptides and surrounds the lipid bilayer cytoplasmic membrane [2]. In Gram-negative bacteria with an outer membrane, such as *Escherichia coli*, PG forms a thin layer, while in Gram-positive bacteria without an external membrane, such as *Bacillus subtilis*, it forms a thick layer. Throughout a bacterium’s life, PG is constantly synthesized, remodeled and repaired to enable cell elongation and division [1,3,4]. The newly synthesized PG is integrated into the existing layer, while the old PG is simultaneously released. These processes are mediated by enzymes such as PG synthases and hydrolases, which belong to dynamic complexes that are spatially and temporally regulated [3,5].

The various steps Involved in PG synthesis have been described in numerous studies and reviews, and this introduction highlights the main ones (Figure 1) [1,6]. The initial stages of PG synthesis occur in the cytoplasm [7]. Uridine diphosphate N-acetylglucosamine (UDP-GlcNAc), a precursor of PG, is synthesized from fructose-6-phosphate via the hexosamine pathway [8,9]. MurA and MurB enzymes catalyze the formation of UDP-N-acetylmuramic acid (MurNAc) from UDP-GlcNAc and phosphoenolpyruvate [10,11]. Five amino acids, including D-amino acids, are then successively added to form UDP-MurNAc-pentapeptide, which varies in its composition depending on the bacteria. An L-alanine (L-Ala) is usually found in the first position. Then, a D-glutamic acid (D-Glu) follows; it is sometimes amidated in Gram-positive bacteria, yielding a D-glutamine (D-Gln) [6]. The γ-carbon of this residue is connected to a third amino acid that is the most variable in the pentapeptide composition. It is usually a dibasic meso-diaminopimelic acid (m-A_2_pm), but in most Gram-positive bacteria, it is usually an L-lysine (L-Lys). Finally, the peptide stem ends with two D-Ala. The composition of the pentapeptide is similar in *E. coli* [12] and in *B. subtilis* [13] and is L-Ala–D-Glu–m-A_2_pm–D-Ala–D-Ala.

The subsequent steps of PG assembly take place on the inner face of the cytoplasmic membrane with enzymes bound to the membrane. In both Gram-positive and Gram-negative bacteria, MraY catalyzes the initial membrane step, forming undecaprenyl-phosphate-N-acetylmuramyl-pentapeptide (Lipid-I) from UDP-MurNAc-pentapeptide and undecaprenyl-phosphate (UndP), a C55 polyisoprenoid lipid phosphate, [14,15,16]. The final intracellular step involves the transfer of a molecule of UDP-GlcNAc to the MurNAc of Lipid-I under the action of MurG, forming Lipid-II (UndPP-GlcNAc-MurNAc-pentapeptide) [17]. Lipid-II is then translocated across the cytoplasmic membrane by the flippase MurJ to the periplasm (Gram-negative) or the exterior (Gram-positive) [18]. The disaccharide-pentapeptide part of Lipid-II serves as a substrate for PG synthases, i.e., PBPs and Shape, Elongation, Division and Sporulation (SEDS) proteins, for polymerization and cross-linking to the PG layer, also named sacculus. Finally, the remaining undecaprenyl pyrophosphate (UndPP) is then dephosphorylated by membrane phosphatases to be transported and recycled as UndP to the cytoplasm [19,20].

PG synthesis from disaccharide pentapeptide is carried out by two distinct PG synthetic machineries: the elongasome and the divisome, which are specific to lateral synthesis (cell elongation) and septal synthesis (cell division), respectively [2,21]. In both cases, two reactions are required: transglycosylation of the disaccharides to form glycan strands and transpeptidation of the peptide stems to form cross-bridges. However, the existence of these two machineries in ovoid bacteria is debated. The elongasome, which seems to be present only in rod-shaped bacteria, is proposed to direct the lateral insertion of PG into the cell along the long axis, enabling cylindrical growth [21]. The two model rod-shaped bacteria, *E. coli* and *B. subtilis*, encode several glycosyltransferase (GTase) and transpeptidase (TPase) enzymes responsible for the synthesis of sidewall and septum PG [1]. Among these enzymes, PBPs are well-characterized and are the targets of several antibiotics [22,23]. PBPs can be subdivided according to their molecular weight (MW), with high-MW PBPs forming two classes: class A (aPBPs), exhibiting both GTase and TPase activities, and class B (bPBPs), carrying only TPase activity. Meanwhile, low-MW PBPs have D,D-carboxypeptidase or endopeptidase activity [1,2,24]. PBPs act in collaboration with the essential proteins RodA and FtsW, which belong to the SEDS protein family. RodA and FtsW also exhibit GTase activity, enabling, respectively, the elongation and division of the bacterial cell [25,26,27]. Extracellular enzymes with autolytic activities are also required for PG expansion by breaking bonds in pre-existing material [3,5,28]. For example, the conserved lytic GTase MltG was proposed to act as a terminase for both aPBP and SEDS/bPBP by cleaving PG glycans as they are actively synthesized [29,30].

The purpose of this mini-review is to highlight recent data concerning the synthesis and regulation of PG during bacterial growth and division. We will focus on specific steps and the respective roles of PBPs and SEDS, mainly in the two model rod-shaped bacteria, the Gram-negative *E. coli* and the Gram-positive *B. subtilis*.

## 2. Regulation of GlcN-6-P Synthesis, the Initial Cytoplasmic Precursor of UDP-GlcNAc

The synthesis of PG begins in the cytoplasm with the formation of UDP-GlcNAc (Figure 1). The UDP-GlcNAc biosynthesis pathway involves four steps and three enzymes: GlmS, GlmM and GlmU, first identified in *E. coli* [31]. GlmM and GlmU are essential. In contrast, GlmS is required only in the absence of amino sugars in the environment. These sugars can be incorporated and converted into glucosamine-6-phosphate (GlcN-6-P), thus bypassing the GlmS-catalyzed reaction. The first, rate-limiting step is the conversion of fructose-6-phosphate (F6P) into GlcN-6-P in the presence of glutamine by GlmS. This step was shown to be highly feedback-regulated in both *E. coli* and *B. subtilis*, although the studies used different molecular mechanisms [32,33,34,35].

### 2.1. Regulation of GlmS Synthesis in E. coli

In *E. coli*, GlmS synthesis is regulated by the GlcN-6-P intracellular level via a mechanism based on four main actors: the two small RNAs (sRNAs), GlmZ and GlmY, the RNase adaptor RapZ and the endoribonuclease RNaseE [36] (Figure 2A,B).

Indeed, the expression of *glmS* is activated by an sRNA, GlmZ [33]. When the intracellular GlcN-6-P concentration is low (Figure 2A), the RapZ protein interacts with the two-component system QseE/QseF to boost the expression of another sRNA, GlmY. GlmY interacts with RapZ [37,38], acting as an anti-adaptor, sequestering RapZ into stable complexes through an RNA mimicry mechanism. This RapZ sequestration protects the sRNA GlmZ from degradation by RNaseE and thus indirectly activates GlmS synthesis [37]. Indeed, GlmZ accumulates and activates the *glmS* mRNA through base-pairing [36,39]. Binding to Hfq facilitates GlmZ base-pairing with the *glmS* mRNA, which stimulates its translation, thus leading to high GlmS enzyme levels and GlcN-6-P replenishment. When the intracellular GlcN-6-P concentration is high (Figure 2B), the two-component system QseE/QseF does not boost the expression of GlmY. This sRNA is rapidly degraded and can no longer sequestrate RapZ. RapZ binds to GlcN-6-P. It interacts with the sRNA GlmZ to act as a RNase adaptor and induce its degradation by the endoribonuclease RNase E [40]. The RNase E cleaves GlmZ at the base-pairing site and thus prevents the stimulation of *glmS* translation, leading to a basal enzyme level [36].

### 2.2. Regulation of Synthesis and Activity of GlmS in B. subtilis

In *B. subtilis*, both GlmS synthesis and activity are feedback-regulated by the GlcN-6-P and UDP-GlcNAc intracellular pools, respectively (Figure 2C,D).

In this Gram-positive bacterium, the level of *glmS* transcript is also controlled according to the GlcN-6-P intracellular concentration. The discovery of this well-studied regulation highlighted a new class of ribozymes [32,41,42]. In fact, the *glmS* transcript contains a unique type of riboswitch in its 5′ UTR: a self-cleaving ribozyme activated by GlcN-6-P. At a low GlcN-6-P concentration (Figure 2C), the already fully folded ribozyme is not complexed to GlcN-6-P; the *glmS* transcript is stable and translated. At high GlcN-6-P concentrations (Figure 2D), this amino sugar binds to the ribozyme, acting as a coenzyme to stimulate RNA self-cleavage [43,44]. No longer protected by a 5′ triphosphate, the *glmS* transcript undergoes rapid exonucleolytic degradation by RNase J1; thus, the GlmS concentration is reduced.

An additional level of regulation of GlcN-6-P synthesis was demonstrated via the feedback regulation of GlmS activity by UDP-GlcNAc [34,45]. In fact, GlmS activity was shown to be stimulated by GlmR, a UDP-GlcNAc-binding protein required for growth on non-glycolytic carbon sources [46,47]. At a low UDP-GlcNAc concentration, i.e., growth on non-glycolytic carbon sources, GlmR is not complexed with UDP-GlcNAc and binds to GlmS in order to stimulate its activity (Figure 2C). In such conditions, GlmR is essential. This stimulation is prevented by the presence of UDP-GlcNAc in vitro or when growth conditions are favorable in vivo, i.e., glycolytic carbon sources (Figure 2D) [34,45]. In such conditions, GlmR is dispensable; it does not interact with GlmS but is complexed with UDP-GlcNAc to bind to YvcJ, a protein homologous to RapZ [34]. In *B. subtilis*, the *glmS* transcript level is not modified by the deletion of *yvcJ*, and YvcJ does not regulate the synthesis of GlmS; its molecular role is not characterized [35]. There is no evidence to show that YvcJ is an RNA-binding protein, with the RNA-binding domain of *E. coli* RapZ being weakly conserved in *B. subtilis* YvcJ [40]. However, YvcJ function is directly or indirectly related to natural competence. Indeed, competence efficiency is affected in *yvcJ* mutant strains in comparison to WT strains, and the expression of ComK regulon is also affected [35,48]. A recent publication confirmed the central role of GlmR in PG synthesis and antibiotic sensitivity [49]. In fact, mutations of *rpoB*, encoding the β-subunit of the RNA polymerase, can alter *B. subtilis* sensitivity to antibiotics such as rifampicin and/or β-lactam cefuroxime. The cefuroxime induces the accumulation of UDP-GlcNAc, which feedback regulates GlmS activity in a GlmR-dependent manner and thus PG synthesis.

## 3. Flipping over the Cytoplasmic Membrane

For decades, the mechanism behind the flipping and recycling of Lipid-II across the cytoplasmic membrane remained unknown. However, two recent publications shed light on this mystery through the simultaneous discovery of two protein families capable of recycling UndP [19,20].

### 3.1. The Lipid-II Flippases

Lipid-II flippases, such as MurJ, are responsible for flipping Lipid-II to the outer leaf of the cytoplasmic membrane (Figure 1) [18]. This protein requires proton motive force (PMF) in order to drive conformational changes for the flip of Lipid-II [50,51]. In *E. coli*, MurJ is the sole Lipid-II flippase. It is essential, and the depletion of this protein results in the inhibition of PG biosynthesis and accumulation of lipid-linked PG precursors [18]. By contrast, *B. subtilis* has four MurJ homologs. One of them (SpoVB) is required for PG synthesis during sporulation [52]. *E. coli* MurJ has been shown to be able to complement the sporulation defect of a *spoVB* mutant. In turn, both SpoVB and YtgP, another MurJ from *B. subtilis*, were shown to complement the growth defect of an *E. coli* strain depleted of *murJ*, thus confirming their status as Lipid-II flippases. Unexpectedly, a *B. subtilis* strain lacking these four MurJ homologs did not exhibit a growth defect, suggesting the existence of an alternative Lipid-II flippase family. Indeed, an additional Lipid-II flippase, Amj (alternate to MurJ), was discovered in *B. subtilis* a few years ago [53]. Cells lacking both Amj and the four MurJ homologs exhibited cell shape defects and lysis. Furthermore, the expression of *amj* or *murJ* from *B. subtilis* in an *E. coli murJ* mutant restored Lipid-II flipping and, consequently, the viability of this mutant. In addition, the SEDS protein, FtsW [27], an essential protein carrying GTase activity required for glycan strand polymerization during septum formation in cell division, has also been shown to transport Lipid-II in vitro [54]. It may thus be an alternative Lipid-II flippase, but this assumption is controversial. This proposition was based on an in vitro FtsW reconstitution assay that reported both a Lipid-II flippase activity and an ability of FtsW to translocate various phospholipids [54]. However, no FtsW-dependent flippase activity could be detected when MurJ was incorporated into liposomes. In addition, no genetic evidence supports the argument that FtsW flips Lipid-II. In fact, it was shown that FtsW activity and Lipid-II synthesis are required for the recruitment of MurJ to the mid-cell in *E. coli* [55]. The mid-cell is the place where septal PG is synthesized and thus where the Lipid-II is flipped during cell division. This result strongly suggests that MurJ and FtsW work together in vivo for the flipping of Lipid-II.

### 3.2. The UndP Flippases

The step following the flipping of Lipid-II to the outer face of the cytoplasmic membrane, where the muropeptide is polymerized and crosslinked to the existing PG meshwork, is the dephosphorylation of UndPP by UndPP membrane phosphatases (BacA, YbjG, PgpB and LpxT in *E. coli* [56] or UppP and BcrC in *B. subtilis* [57]) for its recycling (Figure 1). The resulting UndP is flipped back to the inner side of the cytoplasmic membrane to be recycled for the production of novel Lipid-II. For years, the transporters involved in this recycling were unidentified. It was only very recently that, using genetic screens in *B. subtilis*, *Staphylococcus aureus* [19] and *Vibrio cholerae* [20], two broadly conserved families of flippases were shown to be responsible for UndP transport across the membrane. Genetic, cytological and syntenic analyses support the idea that these two UndP transporter families (corresponding to UptA and PopT proteins) are indeed UndP flippases.

UptA (for UndP transporter A) is a member of the DedA superfamily, a broadly conserved but poorly characterized membrane protein family [19,20]. The UptA structural model resembles the structure of membrane transporters (Figure 3A). Bioinformatic analysis revealed that *B. subtilis* possesses six paralogs of DedA, whereas the *E. coli* genome encodes eight of them [19]. In addition to UptA, *S. aureus* and *V. cholerae* possess another protein involved in UndP recycling, named PopT (polyprenyl-phosphate transporter). This protein is absent in *B. subtilis* and in *E. coli* [19,20]. It carries a DUF368 domain and belongs to a protein family distinct from DedA. Its structural model resembles the structure of canonical membrane transporters, with a two-fold inverted symmetry (Figure 3B). The two types of transporters, UptA and PopT, possess two membrane re-entrant loops into the lipid bilayer (Figure 3). The importance and respective roles of UptA- and PopT-like proteins have not yet been clearly characterized in bacteria that possess these two types of transporters, such as *S. aureus* and *V. cholerae*. However, it was proposed that both transport and recycle UndP. To support this proposition, it was proposed that UndP recycling is reduced in a *B. subtilis* mutant strain lacking YngC, one of the six DedA proteins [19], and in an *S. aureus* mutant strain lacking DUF368-containing proteins [20]. In addition, an *S. aureus* mutant with both *uptA* and *popT* deleted for is highly susceptible to tunicamycin, consistent with reduced levels of inward-facing UndP due to an accumulation of outward-facing UndP in this double mutant [19]. Moreover, members of these two protein families were found in several Gram-positive and Gram-negative bacteria, indicating that the UndP recycling mediated by these flippases is broadly conserved [19,20]. Nevertheless, the mechanisms of UndP transport by these proteins remain unknown and will probably be the subject of future studies.

## 4. Regulation of PG Expansion

After Lipid-II flips to the outer side of the cytoplasmic membrane, GTases polymerize GlcNAc-MurNAc-pentapeptide moieties into glycan strands (Figure 1). TPases then crosslink these strands to form the newly synthesized PG. PBPs (from classes A and B) and SEDS proteins catalyze these reactions [3]. Widely used antibiotics target PG synthesis, highlighting its essential function in bacteria. Rod-shaped bacteria have two modes of PG synthesis during the cell cycle: the elongasome and the divisome. The elongasome catalyzes lateral PG synthesis during cell elongation, while the divisome catalyzes septal PG synthesis during cell division [1,2,21]. The composition of these cell wall machineries varies according to the bacteria. In rod-shaped bacteria, the elongasome is associated with the actin-like protein MreB. This cytoskeletal protein polymerizes into small filaments that localize in patches on the inner face of the lateral cell wall of bacteria in order to drive the elongasome and thus the insertion of new PG during elongation [58,59]. In addition, *B. subtilis* encodes three actin-like proteins of the MreB family (MreB, Mbl and MreBH), MreBH being essential for the activation of the major autolysin LytE and for its localization to the sites of new PG insertion [28,60]. The elongasome, also called Rod-complex, is composed of a variety of enzymes, some of which are found in the composition of the core of this machinery in *E. coli* and *B. subtilis*, such as MreB, MreC, MreD, RodA, RodZ and bPBPs [1,2,60,61]. RodZ physically links the cytoplasmic MreB to the elongasome. aPBPs are not considered as components of this elongasome.

Cell division is controlled by another cytoskeletal protein, FtsZ, a tubulin-like protein present in almost all bacteria. FtsZ polymerizes in the mid-cell to form a circumferential ring, called the Z-ring, that defines the site of division, named the septum [62,63]. FtsZ also recruits numerous proteins to form the divisome [2,64]. The divisome is more complex than the elongasome. In *E. coli*, approximately 40 proteins were identified in its composition. Among them, a dozen are essential or conditionally essential and highly conserved in bacteria. The proteins FtsZ, FtsA, ZipA, FtsE, FtsX, FtsK, FtsQ, FtsL, FtsB, FtsW, FtsI and FtsN constitute the basic components of the divisome [65]. Some of these proteins are also part of the divisome of *B. subtilis*, which is assembled in at least two distinct steps [64]. The first step involves the polymerization of FtsZ and the concomitant recruitment of “early” divisome proteins including FtsA, SepF, ZapA and EzrA. Then, a second step takes place with the recruitment of the “late” proteins that have extracellular or membrane domains and the regulatory proteins, such as GpsB, DivIVA, FtsL, DivIB, FtsW, PBP2b, MinJ, MinD and MinC. The divisome is responsible for both the constriction of the inner (and outer) membrane(s) and PG synthesis at the division site [21] and is almost ubiquitous.

Because PG plays a fundamental role in bacteria, the enzymes involved in its synthesis, remodeling and repair are expected to be highly controlled in a cell-cycle-dependent manner, particularly their synthesis, localization and activity. For example, cell wall elongation and cell division machineries are in competition, and the rates of side wall and septal PG synthesis are inversely correlated [66]. This observation strongly supports the notion of antagonist activities of the elongasome and the divisome, as well as the spatiotemporal regulation of these machineries. Although the way in which SEDS proteins are regulated remains unknown, many regulatory proteins have been found to regulate the localization or activity of PBPs and hydrolases.

### 4.1. Regulation of PBPs

*E. coli* possesses 12 PBPs, which include 3 aPBPs (PBP1a, PBP1b and PBP1c), 2 bPBPs (PBP2 and PBP3) and 7 low-MW PBPs. PBP1a and PBP1b are the major aPBPs, whereas PBP2 is involved in cell elongation and PBP3 belongs to the divisome [67]. The seven low-MW PBPs are involved in cell separation, PG maturation and recycling. *B. subtilis* encodes 16 PBPs, including 4 aPBPs (PBP1a, PBP2c, PBP4 and PBP1d) and 6 bPBPs (PBP2a, PBP2b, PBP3, SpoVD, PBPH and YrrR) [24,68]. The main aPBP is PBP1a. PBP2a and PBPH are involved in cell elongation, with the simultaneous deletion of the two corresponding genes being lethal [69]. PBP2b belongs to the divisome [70]. Some of these PBPs are functionally redundant [68,71]. The production and the properties of these enzymes are relatively well-characterized, but the regulation of their activity and their dynamic localization are not entirely understood. For example, the cell division protein GpsB, which was first proposed to be involved in the shuttling of PBP1a, the main *B. subtilis* aPBP, through the facilitation of its removal from the cell pole [72], was recently shown to have a more complex role. It was proposed to mediate the interaction between PBP1a and various proteins to form larger protein complexes at specific sites, probably to drive PG synthesis in a bacterial cell-cycle-dependent manner [73]. In *E. coli*, two outer-membrane lipoproteins, LpoA and LpoB, were shown to regulate PBP activities and be essential aPBP cofactors [74,75]. LpoA interacts with its cognate PBP, PBP1a, whereas LpoB specifically interacts with PBP1b. LpoA stimulates TPase activity of PBP1a in vitro [76]. In fact, LpoA not only modulates PG crosslinking but is also required for PBP1a to form PG glycan strands. Concerning LpoB, it regulates the PG synthesis rate by spanning the periplasm and reaching its cognate PG synthase, PBP1b [77]. In *B. subtilis*, no essential regulator of PBPs has been highlighted thus far, but the non-essential TseB protein was proposed to regulate PBP2a, a bPBP [78]. Indeed, this membrane protein was shown to be required for efficient cell wall elongation and to specifically interact with PBP2a. However, TseB is not required for PBP2a activity or localization, although PBP2a overproduction is deleterious in the absence of TseB. It was proposed to be a component of the elongasome, regulating PBP2a and cell elongation. In *Streptococcus pneumoniae* and *S. aureus*, it was proposed that MacP and the CozE proteins can act as aPBP regulators [79,80,81] and that EloR can regulate the lytic pneumococcus GTase MltG in the mid-cell [82,83].

### 4.2. Regulation of PG Hydrolases

In addition to PG synthase, PG hydrolases are essential for cell wall elongation and daughter cell separation during division [1]. In the rod-shaped bacteria *E. coli* and *B. subtilis*, cell elongation requires the cleavage of peptide crosslinks within the existing PG via endopeptidases to create space for the incorporation of newly synthetized PG into the cell cylinder. These enzymes are thus required for PG expansion during cell growth. Despite their importance, the regulation of these enzymes is not well-understood (see [5] for a recent review).

*B. subtilis* encodes two functionally redundant D,L-endopeptidases (CwlO and LytE) that, together, are essential [84]. They act on the lateral PG, cleaving peptide crosslinks. Their activity is highly regulated. One aspect of this regulation involves the essential WalR–WalK two-component system. It was shown that WalR–WalK senses the cleavage products generated by CwlO and LytE and, in response, modulates their hydrolase activity during cell wall elongation [85]. The hydrolase activity of CwlO is also regulated by FtsEX, a conserved ABC transporter, with its essential co-factors SweD and SweC, required for cell wall elongation [86].

In *E. coli*, FtsEX plays a role in cell division [87], specifically in the assembly and activation of the divisome so as to connect cell wall synthesis to cell wall hydrolysis at the septum. The FtsEX complex is recruited to the Z-ring through FtsE interaction with FtsZ and then interacts with FtsA to initiate the assembly of the divisome complex while halting septal PG synthesis until the divisome assembly is complete [87,88].

## 5. Respective Role of PBPs and SEDS

It was long believed that aPBPs, which are essential in most bacteria [89], are the only GTases involved in PG synthesis. However, a *B. subtilis* mutant lacking the four aPBPs was shown to be viable and to produce a PG exhibiting an almost normal structure [90], which contradicted this assumption. The discovery of the SEDS proteins RodA and FtsW [26,27] explained the viability of this quadruple mutant. It was proposed that in bacteria with a cell wall, aPBPs are essential, excepting cases where SEDS proteins can replace them [91]. Indeed, RodA and FtsW were shown to be the main PG synthases and to serve as primary GTases [27]. FtsW seems to be universally essential, but some bacteria do not require RodA for growth. In *E. coli* and *B. subtilis*, RodA and FtsW are both essential [92,93]. They interact with their bPBP partners and are the core constituents of the elongasome and the divisome, respectively [26,92]. RodA is necessary for lateral PG synthesis during elongation and for the maintenance of the rod shape of *B. subtilis*. Conditional mutants depleted of either RodA or its two associated bPBPs, PBP2a and PBPH, have a spherical morphology [69,94]. RodA serves as a GTase, while the two redundant bPBPs, PBP2a and PBPH, and possibly aPBPs, provide TPase activity [26,27]. In addition, structural data suggest that these associated bPBPs act as allosteric activators of RodA, permitting the SEDS/bPBP complex to coordinate its double enzymatic activities of PG polymerization and crosslinking in order to build the cell wall [25].

Concerning FtsW, it is an essential component of the divisome and is responsible for septal PG synthesis during cell division, together with its cognate class bPBP (PBP3 in *E. coli*) [95,96]. The precise role of aPBPs had to be reinvestigated in light of the essentiality of SEDS. It was proposed that the SEDS/bPBPs complex builds the foundational PG structure, while the aPBPs fill in the gaps and play a role in the repair of PG defects [97].

In a *B. subtilis* mutant strain that lacks the main aPBP, PBP1a, the overexpression of *rodA* occurs [60]. Moreover, an artificial increase in *rodA* expression can rescue a mutant lacking PBP1a [26]. These observations suggest that the overproduction of RodA can compensate for the absence of aPBP, triggering alternative mechanisms to restore the normal function of the cell wall in *B. subtilis* [60]. For instance, the alternative RNA polymerase sigma-54 factor, σ^I^, was shown to be vital in the absence of aPBPs. In this mutant, an upregulation of MreBH, a homolog of MreB that localizes the LytE autolysin to the elongasome, occurs in a σI-dependent manner. An equilibrium in MreBH–LytE activity is critical for optimal elongasome function. In the absence of aPBPs, increased levels of RodA, MreBH and LytE stimulate the elongasome to enhance lateral PG synthesis, compensating for the loss of mechanical stability of the PG conferred by aPBPs [60]. Thus, aPBPs do not contribute to cell shape determination but are responsible for the mechanical stability of PG, which can be restored through the overproduction or stimulation of the elongasome (particularly RodA).

A study on the respective roles of RodA, the elongasome and aPBPs in cell growth and width demonstrated that increased elongasome activity is correlated with an increased density of directional MreB filaments [98]. The width of *B. subtilis* and *E. coli* rods depends on the balance between the activities of the elongasome, which reduces the cell diameter, and aPBPs, which increase it, indicating a mechanical or reparatory function of aPBPs (especially PBP1a). This hypothesis was reinforced by a recent study proposing that the intrinsically disordered domain (IDR) of some membrane proteins can sense gaps in the PG meshwork and play a role in maintaining cell wall homeostasis in *B. subtilis* [99]. Specifically, PBP1a exhibits an extracytoplasmic IDR that is critical for its function. The IDR of PBP1a directs it to gaps in the PG meshwork in order to repair and strengthen it. In these conditions, PBP1a, whose role is to repair the PG meshwork, complements the elongasome, whose role is to build the foundational PG structure [99].

Based on a model proposed for *S. pneumoniae*, in which the mature PG is synthesized by three functional entities, the divisome (with FtsW), the elongasome (with RodA) and bifunctional PBPs (aPBPs) [100], we can suggest that a similar model exists in *E. coli* and *B. subtilis*, with distinct and specific roles for each of these entities in cell division, cell wall elongation, and cell wall repair, respectively (Figure 4).

## 6. Conclusions

In this mini-review, we explored recent progress in our understanding of the synthesis and regulation of peptidoglycan (PG) in model bacteria, including *E. coli* and *B. subtilis*. Our focus was on protein–protein interactions that regulate PG synthases and hydrolases, as well as the newly discovered factors involved in their flipping across the cytoplasmic membrane. We also discussed the respective roles of SEDS proteins and aPBPs. Despite recent advancements, one of the major challenges in this field is to decipher how different enzymes involved in PG synthesis, repair and remodeling interact with one another and are regulated both spatially and temporally during cell elongation and division on a single-cell level. It is likely that future research will expand our knowledge to non-model bacteria, leading to the discovery of additional actors and mechanisms specific to their developmental processes and environments. Overall, this review underscores the importance of ongoing research on PG synthesis and regulation and its potential to enhance our understanding of bacterial biology and adaptation.

## Figures and Tables

**Figure 1 biomolecules-13-00720-f001:**
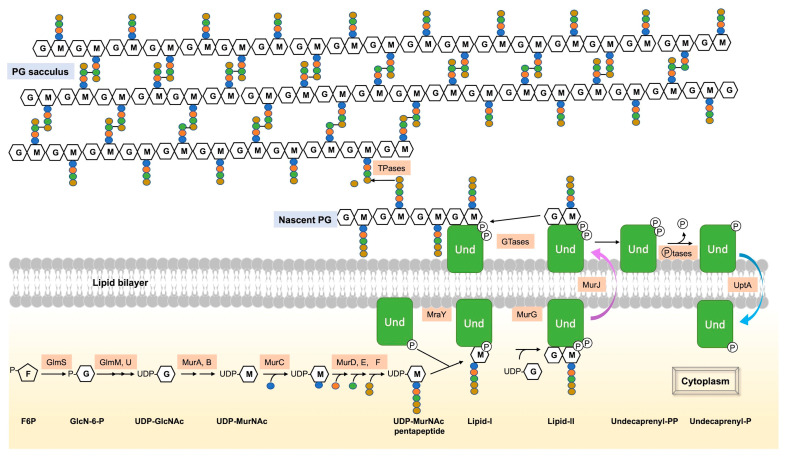
Summary of the main steps of PG synthesis. PG synthesis begins with a series of steps in the cytoplasm. The initial PG precursor, UDP-GlcNAc, is formed from F6P via the hexosamine pathway. UDP-MurNAc is then synthesized from UDP-GlcNAc by the enzymes MurA and MurB. Successive addition of five amino acids forms UDP-MurNAc-pentapeptide. The subsequent steps of PG synthesis take place on the inner face of the cytoplasmic membrane. MraY catalyzes the combination of UDP-MurNAc-pentapeptide with UndP to form Lipid-I, which is then modified by the transfer of UDP-GlcNAc to the MurNAc unit to form Lipid-II by MurG. Flippase MurJ facilitates the translocation of Lipid-II across the cytoplasmic membrane. On the external face of the membrane, GTases and TPases utilize the disaccharide pentapeptide of Lipid-II as a substrate for PG polymerization and synthesis of the PG layer (sacculus) in the periplasm (Gram-negative) or exterior (Gram-positive) of the cell. UndPP, released during PG synthesis, is dephosphorylated by membrane phosphatases (such as BacA, YbjG, PgpB, and LpxT in *E. coli* or UppP and BcrC in *B. subtilis*), and the resulting UndP is recycled to the cytoplasm by the UndP transporter UptA of the DedA superfamily, found in both *B. subtilis* and *E. coli*.

**Figure 2 biomolecules-13-00720-f002:**
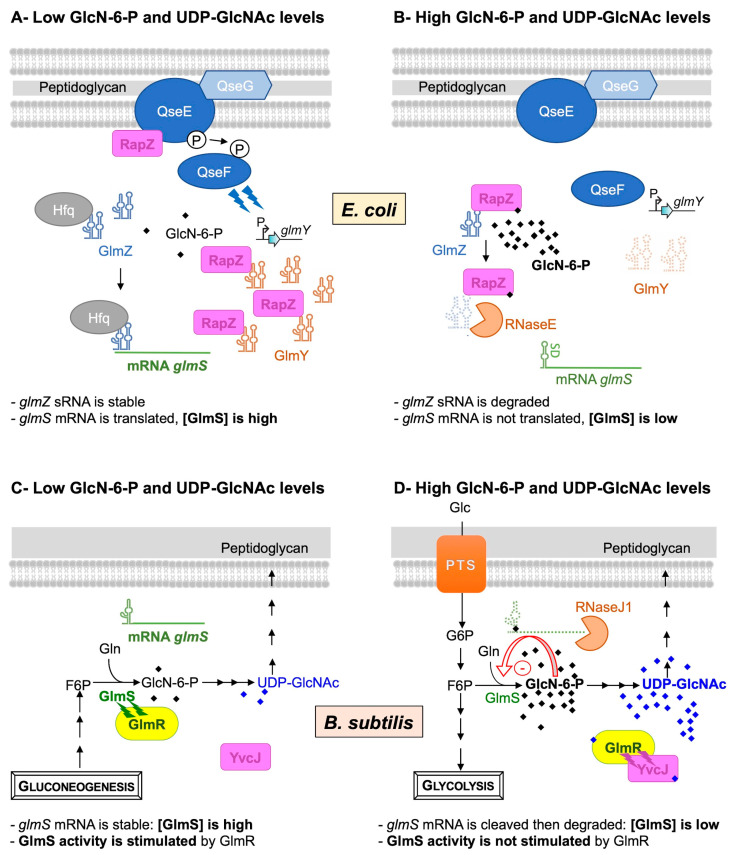
Feedback regulation of GlmS in *E. coli* (**A**,**B**) and in *B. subtilis* (**C**,**D**). In *E. coli*, when the intracellular GlcN6P concentration is low (**A**), the two-component system QseE/QseF, associated with the lipoprotein QseG, boosts the expression of the sRNA GlmY that protects the second sRNA GlmZ from degradation and thus indirectly activates *glmS*. The sRNA GlmZ accumulates and interacts with the *glmS* mRNA through a base-pairing interaction, stabilized by Hfq, to stimulate its translation and increase the GlmS enzyme level. When the intracellular level of GlcN6P is high (**B**), RapZ binds to GlcN6P, thereby interfering with sRNA binding and leading to the stimulation of QseE/QseF. RapZ is released from complexes with GlmY, and the sRNA is rapidly degraded. Once free, RapZ binds and targets GlmZ sRNA to the RNase E endoribonuclease, which cleaves the sRNA at the base-pairing site, thus preventing the stimulation of *glmS* translation. In *B. subtilis*, when intracellular GlcN6P and UDP-GlcNAc concentrations are low (**C**), the ribozyme is not complexed to GlcN-6-P; the *glmS* transcript is stable and translated to increase the GlmS enzyme level. In addition, GlmR interacts with GlmS to stimulate its activity. When the intracellular GlcN6P and UDP-GlcNAc concentrations are high (**D**), GlcN6P binds to the ribozyme of the *glmS* transcript and stimulates its self-cleavage. No longer protected by a 5′ triphosphate end, the *glmS* transcript undergoes rapid exonucleolytic degradation by RNase J to decrease the GlmS enzyme level. In addition, GlmR binds UDP-GlcNAc and no longer interacts with GlmS to stimulate its activity. However, it binds to YvcJ, a protein homologous to RapZ.

**Figure 3 biomolecules-13-00720-f003:**
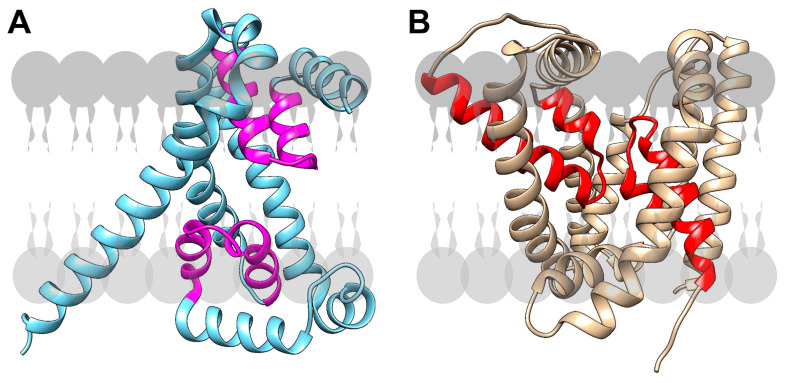
Structural models of the two families of UndP membrane transporters from the AlphaFold Protein Structure Database. (**A**) Structural model of a UndP transporter of the DedA superfamily (UptA from *B. subtilis*). The model of the *B. subtilis* UptA (YngC) transporter has two membrane loops that re-enter the cytoplasmic lipid bilayer (in magenta); they are frequently found in membrane transporters. (**B**) Structural model of UndP transporter of the DUF368 superfamily (PopT from *S. aureus*). The model of *S. aureus* PopT (SAOUHSC_00846) also has two loops that re-enter the cytoplasmic membrane (in red).

**Figure 4 biomolecules-13-00720-f004:**
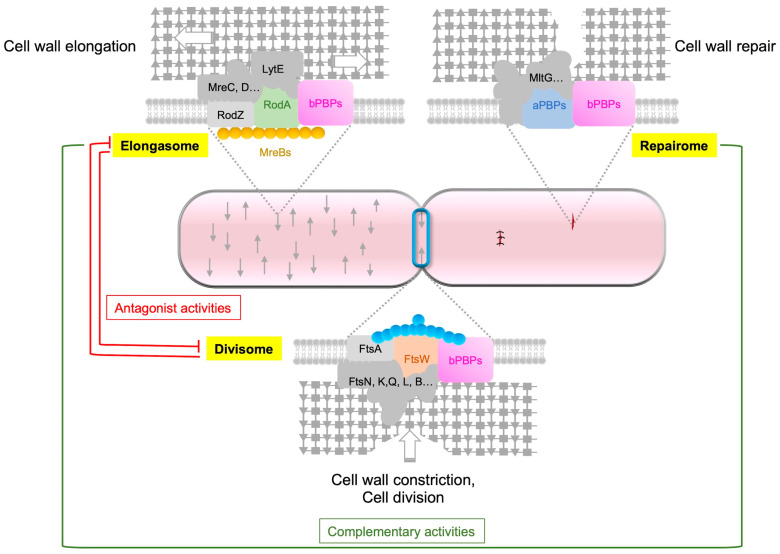
Respective roles of aPBP, the elongasome and the divisome. The synthesis of PG is carried out by three distinct complexes, namely, the elongasome, the divisome and the repairome. The elongasome is responsible for constructing the foundational lateral PG structure, and the GTase RodA plays a key role in this process. The repairome, which includes aPBPs, is involved in repairing and strengthening the PG meshwork in collaboration with the elongasome complex. In *B. subtilis*, PBP1a, the main aPBP, has an extracytoplasmic IDR that guides the GTase to gaps in the PG meshwork for repair. When aPBPs are absent, cells increase the expression of *rodA*, *mreBH* and *lytE* genes to upregulate elongasome activity. The divisome, on the other hand, is responsible for building the septal PG, and during cell division, the GTase FtsW plays a key role. Antagonistic activities are observed between the elongasome and the divisome, with the activity of one being repressed during the activity of the other.

## Data Availability

Not applicable.

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
