# Peer review of "Recent Advances in Peptidoglycan Synthesis and Regulation in Bacteria"

_biomolecules, 2023, doi:10.3390/biom13050720_

Round 1
Reviewer 1 Report
This mini-review by Anne Galinier et al. outlines recent advancements in understanding PG synthesis and regulation in the model bacteria E. coli and B. subtilis. The authors chose to focus on protein-protein interactions, proteins that facilitate flipping across the cytoplasmic membrane, PBP proteins, and Shape/Elongation/Division/Sporulation proteins. The manuscript is clearly written and presents up-to-date information. Overall, the quality of the manuscript is good.
The following are the relevant concerns that need to be addressed:
Line 50. The general pentapeptide formulas are unclear. If Glu is amidated in G+ bacteria, then it should be indicated in the general formula. Also, the pentapeptide composition may differ depending on the bacterial species. Therefore, it would be appropriate to mention this or provide only E. coli and B. subtilis pentapeptide sequences.
Additionally, the statement that G+ bacteria do not have mDAP in their pentapeptide is incorrect. B. subtilis, for example, does have the acid (https://onlinelibrary.wiley.com/doi/full/10.1111/mmi.13629). Given that the review is about E. coli and B. subtilis, the authors should make the distinction of pentapeptides among bacteria clearer.
Line 53-54. Please verify whether MurNAc-pentapeptide is transferred onto undecaprenyl phosphate or undecaprenyl pyrophosphate. It should be phosphate.
Line 60-61. Please clarify which enzyme decomposes Lipid-II and into what.
Line 64-65. The text does not match the figure. It is said that UndPP is dephosphorylated and transported, but in the figure, UndP is still phosphorylated. Also, information on how UndP becomes UndPP should be included.
Line 72. In the text, the authors say that the phospho-pentapeptide moiety is transferred onto the pyrophosphate, but here - on phosphate. Please correct it.
Line 111. The word "absolutely" is informal here. Please fix.
Lines 116-117. A citation is needed.
Line 141. Please change "weak" to "low".
Lines 177-186. It is not clear what the message is here. The part begins with YvcJ, then the authors move to GlmR and ComK. This part should be rewritten to be clearer.
Lines 197-198. A citation is needed.
Lines 237-247. This part needs improvement. From the way it is written, I get the impression that DUF368-containing proteins in S. aureus are the only ones responsible for UndP recycling. If yes, then what is the purpose of UptA? Also, is there any information regarding the requirement of UptA and PopT paralogs for E. coli and/or B. subtilis viability?
Chapter 4. Are there any differences regarding the composition of divisome and elongosome between E. coli and B. subtilis? If yes, then it should be clearly indicated. Also, what is the function of monofunctional murein GTases in the regulation of PG expansion? The authors should clearly indicate which proteins belong to the latter family, as it is hard to follow.
Line 304. The authors should include a short description of what other PBPs are and what their functions are.
Author Response
Line 50. The general pentapeptide formulas are unclear. If Glu is amidated in G+ bacteria, then it should be indicated in the general formula. Also, the pentapeptide composition may differ depending on the bacterial species. Therefore, it would be appropriate to mention this or provide only E. coli and B. subtilis pentapeptide sequences. Additionally, the statement that G+ bacteria do not have mDAP in their pentapeptide is incorrect. B. subtilis, for example, does have the acid (https://onlinelibrary.wiley.com/doi/full/10.1111/mmi.13629). Given that the review is about E. coli and B. subtilis, the authors should make the distinction of pentapeptides among bacteria clearer.
This part has been rewritten
Line 53-54. Please verify whether MurNAc-pentapeptide is transferred onto undecaprenyl phosphate or undecaprenyl pyrophosphate. It should be phosphate.
It is undecaprenyl phosphate: It is now corrected
Line 60-61. Please clarify which enzyme decomposes Lipid-II and into what.Line 64-65. The text does not match the figure. It is said that UndPP is dephosphorylated and transported, but in the figure, UndP is still phosphorylated. Also, information on how UndP becomes UndPP should be included.
This part has been rewritten and the figure modified accordingly
Line 72. In the text, the authors say that the phospho-pentapeptide moiety is transferred onto the pyrophosphate, but here - on phosphate. Please correct it.
Corrected
Line 111. The word "absolutely" is informal here. Please fix.
“Absolutely” has been removed
Lines 116-117. A citation is needed
Citation added
Line 141. Please change "weak" to "low".
Change made
Lines 177-186. It is not clear what the message is here. The part begins with YvcJ, then the authors move to GlmR and ComK. This part should be rewritten to be clearer.
This part has been rewritten
Lines 197-198. A citation is needed.
Citation added
Lines 237-247. This part needs improvement.
From the way it is written, I get the impression that DUF368-containing proteins in S. aureus are the only ones responsible for UndP recycling.
No, it is not the case, both transporters seem required.
If yes, then what is the purpose of UptA? Also, is there any information regarding the requirement of UptA and PopT paralogs for E. coli and/or B. subtilis viability?
This part was modified in order to improve it and to answer to these questions.
Chapter 4. Are there any differences regarding the composition of divisome and elongosome between E. coli and B. subtilis? If yes, then it should be clearly indicated.
It is now indicated.
Also, what is the function of monofunctional murein GTases in the regulation of PG expansion? The authors should clearly indicate which proteins belong to the latter family, as it is hard to follow.
We removed the “murein GTases” that are not really involved in regulation of PG expansion.
Line 304. The authors should include a short description of what other PBPs are and what their functions are.
It is now included in the revised version.

Reviewer 2 Report
The review paper by Galinier et al describes the present-day findings on peptidoglycan synthesis and regulation in bacteria. The authors summarize the results of studies carried out in recent years, including those obtained by the authors of this review themselves. Undoubtedly, the review will be very useful for those who are interested both in mechanisms of peptidoglycan synthesis and the general questions of biology. The publication is fully justified.
Few points require attention before acceptance for publication.
I propose to give a clearer interpretation of the abbreviations used (for example, the word "GTase" – "glycosyltransferase" - is not deciphered anywhere).
Lines 257-259: you need to give a link to the publication.
Author Response
The review paper by Galinier et al describes the present-day findings on peptidoglycan synthesis and regulation in bacteria. The authors summarize the results of studies carried out in recent years, including those obtained by the authors of this review themselves. Undoubtedly, the review will be very useful for those who are interested both in mechanisms of peptidoglycan synthesis and the general questions of biology. The publication is fully justified.
Few points require attention before acceptance for publication.
I propose to give a clearer interpretation of the abbreviations used (for example, the word "GTase" – "glycosyltransferase" - is not deciphered anywhere).
Modifications done
Lines 257-259: you need to give a link to the publication.
Citation added

Reviewer 3 Report
In this review article by Galinier et al., the recent advances on peptidoglycan synthesis in bacteria are well documented. Especially, the authors focused on regulation of peptidoglycan synthesis in Escherichia coli and Bacillus subtilis.
Unfortunately, there are so many mistakes in the manuscript. English should be checked by a professional editor.
Line 25. “Ernst Boris Chain et Howard Walter Florey” -> “Ernst Boris Chain and Howard Walter Florey”
Line 29. “a essential” -> “an essential”.
Lines, 48. “l-Ala-γ-d-Glu-mDAP-d-Ala-d-Ala” -> “L-Ala-γ-D-Glu-mDAP-D-Ala-D-Ala”
and so on.
Several references seem not appropriate. The earliest report should be cited.
For example, Ikeda et al., J. Bacteriol., 1991, 173:1021-1026 should be cited for reference 11.
Author Response
In this review article by Galinier et al., the recent advances on peptidoglycan synthesis in bacteria are well documented. Especially, the authors focused on regulation of peptidoglycan synthesis in Escherichia coli and Bacillus subtilis. Unfortunately, there are so many mistakes in the manuscript. English should be checked by a professional editor.
The manuscript was read and corrected in order to remove our French touch.
Line 25. “Ernst Boris Chain et Howard Walter Florey” -> “Ernst Boris Chain and Howard Walter Florey”
Corrected
Line 29. “a essential” -> “an essential”.
Corrected
Lines, 48. “l-Ala-γ-d-Glu-mDAP-d-Ala-d-Ala” -> “L-Ala-γ-D-Glu-mDAP-D-Ala-D-Ala”
Corrected
and so on.
Several references seem not appropriate. The earliest report should be cited.
We added / corrected some references to cite the earliest study.
For example, Ikeda et al., J. Bacteriol., 1991, 173:1021-1026 should be cited for reference 11.
Corrected
